# Efficient Reinforcement Learning for 3D Jumping Monopods

**DOI:** 10.3390/s24154981

**Published:** 2024-08-01

**Authors:** Riccardo Bussola, Michele Focchi, Andrea Del Prete, Daniele Fontanelli, Luigi Palopoli

**Affiliations:** 1Dipartimento di Ingegneria and Scienza Dell’Informazione (DISI), University of Trento, 38123 Trento, Italy; riccardobussola00@gmail.com (R.B.); luigi.palopoli@unitn.it (L.P.); 2Dipartimento di Ingegneria Industriale (DII), University of Trento, 38123 Trento, Italy; andrea.delprete@unitn.it (A.D.P.); daniele.fontanelli@unitn.it (D.F.)

**Keywords:** aerial motions, reinforcement learning, control, trajectory optimization

## Abstract

We consider a complex control problem: making a monopod accurately reach a target with a single jump. The monopod can jump in any direction at different elevations of the terrain. This is a paradigm for a much larger class of problems, which are extremely challenging and computationally expensive to solve using standard optimization-based techniques. Reinforcement learning (RL) is an interesting alternative, but an end-to-end approach in which the controller must learn everything from scratch can be non-trivial with a sparse-reward task like jumping. Our solution is to guide the learning process within an RL framework leveraging nature-inspired heuristic knowledge. This expedient brings widespread benefits, such as a drastic reduction of learning time, and the ability to learn and compensate for possible errors in the low-level execution of the motion. Our simulation results reveal a clear advantage of our solution against both optimization-based and end-to-end RL approaches.

## 1. Introduction

Legged robots have become a popular technology for navigating unstructured terrains. Control design for this type of system is far from trivial. Remarkable results have been achieved for standard locomotion tasks with continuous foot contact (e.g., walking and trotting [1]). Other tasks, like jumps, have a long aerial phase without any contact with the ground, and are more challenging because even small deviations from the desired trajectory can have a large impact on the landing location and orientation [2]. Additionally, the high accelerations and the constraints involved make jumping very challenging to manage, especially in the context of a real-time application.

This problem has received some attention in the last few years. A line of research has produced heuristic approaches that rely on physical intuition and/or simplified models to be used in designing controllers or planners [3,4]. However, the hand-crafted motion actions produced by these approaches are not guaranteed to be physically implementable. Another common approach is to use full-body numerical optimization [5] that enables handling the numerous constraints associated with legged robots. A remarkable set of aerial motions (jumps onto and off of platforms, spins, flips, barrel rolls) has been produced by MIT Mini Cheetah in [6,7,8] using a centroidal momentum-based non-linear optimization. However, optimization-based approaches for high-dimensional non-linear problems suffer from high computational costs, making them unsuitable for real-time implementation, particularly for online trajectory replanning. Recent advances [5,9] have made significant improvements in the efficiency of Model Predictive Control (MPC) for jumping tasks. However, the price to pay is to introduce some artificial constraints, such as fixing the contact sequence, the time-of-flight, or optimizing the contact timings offline [5]. One limitation of pre-specifying foot contacts, it that this can cause instability problems in the presence of a large contact mismatch [10].

A third set of approaches is based on Reinforcement Learning (RL). The increased computational power and the availability of sophisticated approaches for function approximation have fueled an increasing interest in robot learning. The seminal work of Lillicrap [11] showed that an actor-critic scheme combined with a deep Q network could be successfully applied to learn end-to-end policies for a continuous action domain. Given these results, several groups have then applied RL to quadrupeds for locomotion tasks [12,13,14,15,16], and to in-place hopping legs [17]. As with most model-free approaches, the use of RL requires a massive number of training steps (in the order of millions) to find good solutions [18]. Other approaches [19,20,21] seek to improve the efficiency and robustness of the learning process by combining Trajectory Optimization (TO) with RL: they use the former to generate initial trajectories to bootstrap the exploration of alternatives made by the latter. As a final remark, the efficiency and robustness of the RL learning process are greatly affected by the choice of the action space [22,23]. Some approaches require that the controller directly generates the torques [24], while others suggest that the controller operates in Cartesian or joint space [13,25]. In [19], the authors combined trajectory optimization with deep RL, employing trajectory optimization to provide demonstrations used as a starting point to facilitate initial exploration, thus addressing the issue of local optimality.

Despite the abundance of research in the field, the application of RL to a jumping task has received limited attention, and with good reasons. Indeed, due to the prolonged airborne phase, and the sparse/discrete nature of the reward (i.e., a positive reward is given at the touchdown moment), the problem of jumping is not easily addressed with a conventional RL [26]. There are a few learning methods applied to jumping that require prolonged training periods and/or over-engineered reward shaping [27]. A straightforward implementation of end-to-end RL methods without any structure can lead to disappointing learning performance [28]. This is primarily due to the non-smooth reward landscape created by abrupt contact changes. Among successful RL approaches to jump tasks, Yang et al. [28] utilize a controller to warm start a policy for efficient training (i.e., they learn the residual action) but consider only fixed jump durations. The authors claim that warm-starting the policy reduces the noise in the reward landscape, and determines a better convergence of the training process. Vezzi et al. [29] propose multi-stage learning phases with different levels of refinement; but their goal is to maximize jump height and distance, with little or no consideration for jump accuracy.

### Paper Contribution

We propose an RL framework to produce omnidirectional jump trajectories (from standstill) on elevated terrain. The main features are: 1. a computation time below a few milliseconds that enables the real-time execution of the software at controller rates (i.e., in the order of kHz); 2. a quick learning phase; 3. high levels of accuracy and safety. A faster learning phase has two advantages: (1) lowering the barriers to accessing this technology for users with limited availability of computing power; (2) addressing the environmental concerns connected with the carbon footprint of learning technologies [30].

We compare our approach (that we call Guided Reinforcement Learning (GRL)) with both a baseline TO controller with a fixed jump duration and a “standard” End-to-end Reinforcement Learning (E2E) approach that considers joint references as action space. In the first comparison, GRL achieved better or equal performance while ensuring real-time computation speed. With respect to E2E walking approaches [13,24], we observed a substantial reduction in the number of episodes (without considering parallelization) needed to achieve good learning performance [31]. Instead, an E2E implementation specific for a jump motion (Section 4.2), did not provide satisfactory learning results. This aligns with the outcomes of [28]. To the author’s knowledge, nobody has yet achieved good results learning omni-directional jumps with pure E2E approaches.

The paper is organized as follows: Section 2 presents our GRL approach; Section 3 provides implementation details; in Section 4, we showcase our simulation results compared with state-of-the-art approaches; finally, we draw the conclusions in Section 5.

## 2. Guided Reinforcement Learning for Jumping

### 2.1. Problem Description

Simple notions of bio-mechanics suggest that legged animals execute their jumps in three phases: 1. *thrust*—an explosive extension of the limbs follows an initial compression to gain sufficient momentum for the lift-off and the phase finishes when the foot leaves the ground; 2. *flight*—the body, subject uniquely to gravity, reaches an apex where the vertical Center of Mass (CoM) velocity changes its sign and the posture is adjusted to prepare for landing; 3. *landing*—the body realizes a touch-down, which means that the foot establishes contact with the ground again. For the sake of simplicity, we consider a simplified and yet realistic setting: a monopod robot, whose base link is sustained by passive prismatic joints preventing any change in its orientation (see Section 3.3). In this paper, we focus only on the thrust phase. We assume that a specialized controller manages the landing phase, such as the one proposed by Roscia et al. [2].

The flight phase is governed by the ballistic law. Let $c_{tg}$ be the target location for the CoM of the robot at the end of the jump and let $\left(c_{lo},{\dot{c}}_{lo}\right)$ be the CoM state at lift-off. Since the trajectory is ballistic, after lift-off, the trajectory lies on the vertical plane containing $c_{lo}$ and $c_{tg}$. The set of possible landing CoM positions is a function of $c_{lo}$, ${\dot{c}}_{lo}$:(1)$$\left\{\begin{array}{c} c_{tg,xy}=c_{lo,xy}+{\dot{c}}_{lo,xy}T_{fl}\\ c_{tg,z}=c_{lo,z}+{\dot{c}}_{lo,z}T_{fl}-\frac{1}{2}gT_{fl}^{2} \end{array}\right.$$
where $T_{fl}=\left(c_{tg,xy}-c_{lo,xy}\right)/{\dot{c}}_{lo,xy}$ is the flight time. Given these considerations, we can model the problem of generating the thrust phase in the following terms:

**Problem 1.** 
*Synthesize a thrust phase that produces a lift-off state (i.e., CoM position and velocity) that: 1. satisfies (Equation 1); 2. copes with the potentially adverse conditions posed by the environment (i.e., contact stability, friction constraints); 3. satisfies the physical and actuation constraints.*


Non-linear optimization is frequently used for similar problems. However, it has two important limitations that obstruct its application in our specific case: 1. the computation requirements are very high, complicating both the real-time execution and the use of low-cost embedded hardware, 2. the problem is strongly non-convex, which can lead the solvers to be trapped in local minima, 3. local approaches have better performance in terms of computational effort but the quality of the computed solution is usually dependent on a good initialization.

### 2.2. Overview of the Approach

We use RL to learn optimal trajectories for a jump motion, which a lower-level controller then tracks. Our strategy is based on the following ideas. First, learning is performed in Cartesian space rather than in joint space. This choice allows generalization towards different robot morphologies and configurations [32]. Additionally, it allows for the side execution of a safety filter that discards unfeasible outputs of the RL agent by simple computations (see Section 3.2). Second, while the system is airborne, its final landing position is dictated by simple mechanical laws (ballistic). Therefore, the learning process can focus solely on the thrusting phase. Third, we know from biology [33] that mammals are extremely effective in learning how to walk because of “prior” knowledge of their genetic background. The learning process can thus be guided by an approximate knowledge of what the resulting motion should “look like”. Physical intuition suggests that a jump motion needs a “charging” phase to compress the legs, followed by an extension phase where the CoM is accelerated both upwards and in the jump direction. The “charging” phase allows the exploitation the full range of leg extension for CoM acceleration. Specifically, we parametrize the thrusting trajectory (i.e., from standstill to lift-off) for the CoM with a (3rd order) Bézier curve that is instrumental in obtaining such natural-looking trajectories. The adoption of Bézier curves for learning tasks is not uncommon in the literature [34,35,36]. This heuristic drastically reduces the dimension of the search space and gives a physically meaningful reference for trajectory learning.

Additionally, we will show that the trajectories computed with our approach are very close to the ones obtained with numerical optimization, and the system retains good generalization abilities. By making this choice, we can learn the Bézier parameters using an off-policy Deep Reinforcement Learning (Deep-RL) algorithm, Twin Delayed Deep Deterministic Policy Gradient (TD3), trained to minimize cost functions very similar to those typically used in optimal control.

The Cartesian trajectory generated by our RL agent is translated into joint space via inverse kinematics, and tracked by a low-level joint-space Proportional-Derivative (PD) controller with gravity compensation. This turned out to be more effective than parametrizing the actions in joint space, as in E2E RL (see Section 4.3.3). Indeed, despite E2E approaches demonstrating quite effective learning of walking motions [31], they turn out to struggle to learn tasks with sparse rewards [26], and this is the case with jumping motions.

Our RL pipeline is depicted in Figure 1. The agent state can be in training or inference mode; when in training mode, the Critic Neural Network (NN) and the Actor NN are updated (see dashed lines in the figure) at regular intervals after collecting a sufficient number of rewards (cf. Table 1).

## 3. Learning Framework

The thrust phase is characterized by the lift-off CoM position $c_{lo}$ and velocity ${\dot{c}}_{lo}$, and by the thrust duration $T_{th}\in R$, which is the time to reach the lift-off configuration from the initial state. For GRL, the state of the environment is defined as ($c_{0}$, $c_{tg}$), where $c_{0}\in R^{3}$ is the initial CoM position and $c_{tg}\in R^{3}$ the CoM at the landing location (*target*). The objective of the GRL agent is to find the jump parameters ($c_{lo}$, ${\dot{c}}_{lo}$, $T_{th}$) that minimize the landing position error at touch-down $∥c-c_{tg}∥$ while satisfying the physical constraints. Our jumping scenario can be seen as a high-level planning of a single-step episode where the only action performed leads always to the end state.

### 3.1. The Action Space

The dimension of the action space has a strong impact on the performance. An action with few parameters reduces the exploration space, this reduces the complexity of the mapping, speeding up the learning process. A first way to reduce the complexity of the action space is by expressing $c_{lo}$ and ${\dot{c}}_{lo}$ in spherical coordinates (Equation 2). Because of the peculiar nature of a jump task, the trajectory lies in the plane containing the CoM c and its desired target location $c_{tg}$. Hence, we define the yaw angle φ as the orientation of the jumping plane (where the $c_{tg}-c_{lo}$ vector lies) in the X−Y frame. Because the jump trajectory is constrained to be in the jumping plane, φ remains constant (i.e., $\varphi =\bar{\varphi }$) throughout the flight, and we can further restrict the coordinates to a convex bi-dimensional space:(2)$$\begin{array}{c} \left\{\begin{array}{c} c_{lo,x}=r\;\cos\left(\theta \right)\cos\left(\bar{\varphi }\right)\\ c_{lo,y}=r\;\cos\left(\theta \right)\sin\left(\bar{\varphi }\right)\\ c_{lo,z}=r\;\sin\left(\theta \right) \end{array}\right.\left\{\begin{array}{c} {\dot{c}}_{lo,x}=r_{v}\;\cos\left(\theta_{v}\right)\cos\left(\bar{\varphi }\right)\\ {\dot{c}}_{lo,y}=r_{v}\;\cos\left(\theta_{v}\right)\sin\left(\bar{\varphi }\right)\\ {\dot{c}}_{lo,z}=r_{v}\;\sin\left(\theta_{v}\right) \end{array}\right. \end{array}$$

As shown in Figure 2, the lift-off position vector $c_{lo}$ is identified by: the radius *r* (i.e., the maximum leg extension), the yaw angle $\bar{\varphi }$, and the pitch angle θ. Likewise, the lift-off velocity ${\dot{c}}_{lo}$, is described by its magnitude $r_{v}$, and the pitch angle $\theta_{v}$ with respect to the ground.

Therefore, by using this assumption, we have reduced the dimension of the action space from 7 to 5: $a=\left(r,\theta ,r_{v},\theta_{v},T_{th}\right)\in R^{5}$.

The action space can be further restricted by applying some domain knowledge. The radius *r* has to be smaller than a value $r_{max}$ (0.32 m) to prevent boundary singularity due to leg over-extension, and greater than a value $r_{min}$ (0.25 m) to avoid complete leg retraction. The bounds on the velocity ${\dot{c}}_{lo}$, represented by $r_{v}\in \left[0.1,4\right]$ m/s, and $\theta_{v}\in \left[\frac{\pi }{6},\frac{\pi }{2}\right]$ rad, and the bounds on pitch angle $\theta \in \left[\frac{\pi }{4},\frac{\pi }{2}\right]$ rad are set to rule out jumps that involve excessive foot slippage and useless force effort. Specifically, restricting $\theta_{v,min}$ to be positive ensures a non-negligible vertical component for the velocity, while bounding $\theta_{v}$ to the positive quadrant secures that the lift-off velocity will be oriented “toward” the target. Note that the above restrictions, performed at the level of action design, prevent the agent from exploring trajectories that are physically impossible, reducing the search space without any loss in terms of optimality.

#### Trajectory Parametrization in Cartesian Space

Our strategy to tackle the problem of generating a compression–extension trajectory for the leg to achieve a given lift-off configuration $c_{lo}$ is based on two important choices: 1. restricting the search of the Cartesian space evolution to curves generated by known parametric functions; 2. making the RL agent learn the curve trajectory’s parameters and then finding the joint trajectories through inverse kinematics, with the obvious benefit of reducing the search space and boosting convergence.

The analytical and geometric properties of 3rd order Bézier curves make them a perfect fit for our problem. Bézier curves are a class of parametric functions defined on the same variable (i.e., time) which generate points in the convex hull of the control points. A 3rd order Bézier curve is defined by four control points. In our case, the first and the final points are constrained to be the initial and lift-off CoM positions, respectively. Defining the following Bernstein polynomials:(3)$$\begin{array}{cc} \; & \eta \left(t\right)={\left[\begin{array}{cccc} {\left(1-t\right)}^{3} & 3{\left(1-t\right)}^{2}t & 3\left(1-t\right)t^{2} & t^{3} \end{array}\right]}^{T} \end{array}$$(4)$$\begin{array}{cc} \; & \dot{\eta }\left(t\right)={\left[\begin{array}{ccc} {\left(1-t\right)}^{2} & 2(1-t)t & t^{2} \end{array}\right]}^{T} \end{array}$$

We can compactly write the Bézier curve as function of its $P_{i}\in R^{3}$ control points:(5)$$\begin{array}{c} c=\left[\begin{array}{cccc} P_{0} & P_{1} & P_{2} & P_{3} \end{array}\right]\eta \left(t\right) \end{array}$$

The curve domain is defined only in the normalized time interval: $t\in \left[\begin{array}{c} 0,1 \end{array}\right]$. The derivative of a 3rd degree Bézier curve is itself a Bézier curve of 2nd degree with 3 control points defined as $3(P_{i+1}-P_{i})$. This can be easily proved by computing the time derivative of (Equation 5) and finding that it simplifies to a Bézier polynomial of the 2nd order whose coefficients turn out to be $3(P_{i+1}-P_{i})$. Since we are considering an execution time $T_{exe}\in \left[\begin{array}{c} 0,T_{th} \end{array}\right]$ and $t=\frac{T_{exe}}{T_{th}}$, then the derivative becomes:(6)$$\begin{array}{c} \dot{c}=\frac{1}{T_{th}}\left[\begin{array}{ccc} P_{0}^{\prime } & P_{1}^{\prime } & P_{2}^{\prime } \end{array}\right]\dot{\eta }\left(t\right) \end{array}$$

From the definition of the curve (Equation 5) and its derivative (Equation 6), we can compute the control points $P_{i}$ by setting the boundary conditions of the initial/lift-off CoM position $c_{0}$, $c_{lo}$ and initial/lift-off CoM velocity ${\dot{c}}_{0}$, ${\dot{c}}_{lo}$ in (Equation 7).
(7)$$\left\{\begin{array}{c} \begin{array}{cc} P_{0}^{\prime } & =\frac{3}{T_{th}}\left(P_{1}-P_{0}\right)\\ \; & ={\dot{c}}_{0}=0\\ P_{1}^{\prime } & =\frac{3}{T_{th}}\left(P_{2}-P_{1}\right)\\ P_{2}^{\prime } & =\frac{3}{T_{th}}\left(P_{3}-P_{2}\right)\\ \; & ={\dot{c}}_{lo} \end{array} \end{array}\right.\left\{\begin{array}{c} \begin{array}{cc} P_{0} & =c_{0}\\ P_{1} & =\frac{T_{th}}{3}P_{0}^{\prime }+P_{0}\\ \; & =\frac{T_{th}}{3}{\dot{c}}_{0}+c_{0}\\ P_{2} & =-\frac{T_{th}}{3}P_{2}^{\prime }+P_{3}\\ \; & =-\frac{T_{th}}{3}{\dot{c}}_{lo}+c_{lo}\\ P_{3} & =c_{lo} \end{array} \end{array}\right.$$

### 3.2. A Physically Informative Reward Function

In RL, an appropriate choice of the reward function is key to the outcome. Furthermore, we can use the reward function as a means to inject prior knowledge into the learning process. In our case, the reward function was designed to penalize the violations of the physical constraints while giving a positive reward to the executions that make the robot land in the proximity of the target point. The constraints that must be enforced throughout the whole thrust phase are called *path* constraints. Given the lower and upper limits $\underline{x}$, $\bar{x}$ for each variable *x*, their violation is computed as a cost through the usage of a linear activation function $A(x,\underline{x},\bar{x})$:$$A\left(x,\underline{x},\bar{x}\right)=\left|\min\left(x-\underline{x},0\right)+\max\left(x-\bar{x},0\right)\right|$$

The output of the activation function is zero if the value is in the allowed range, and is the exceeded violation otherwise.

**Physical feasibility check**: Before starting each episode, we perform a sanity check on the action a proposed by the RL policy: if the given CoM vertical velocity is not sufficient to reach the target height, we abort the simulation, returning a high penalty cost $C_{ph}$. This can be computed by obtaining the time to reach the apex $T_{fup}={\dot{c}}_{lo,z}/g$ and substituting it in the ballistic equation:(8)$$\begin{array}{cc} {\bar{c}}_{z}\left(T_{fup}\right) & =c_{lo,z}+{\dot{c}}_{lo,z}T_{fup}+\frac{1}{2}\left(-g\right)T_{fup}^{2} \end{array}$$
This results in ${\bar{c}}_{z}\left(T_{fup}\right)=c_{lo,z}+\frac{1}{2}\frac{{\dot{c}}_{lo,z}^{2}}{g}$, which is the apex elevation. If $c_{tg,z}>{\bar{c}}_{z}\left(T_{fup}\right)$, the episode is aborted. This feasibility check can be employed as an “a posteriori” safety feature in the inference phase, to check if a predicted action will lead to unsafe results. In this case, the action can be aborted early without performing the jump, and high-level strategies could be adopted to relax the jumping requirements (e.g., lower the target height).

**Unilaterality constraint**: In a legged robot, a leg can only push on the ground and not pull. This is because the component of the force F along the contact normal (*Z* for flat terrains) must be positive.

**Friction constraint**: To avoid slippage, the tangential component of the contact force $∥F_{x,y}∥$ is bounded by the foot-terrain friction coefficient μ, i.e., $∥F_{x,y}∥\leq \mu F_{z}$.

**Joint range and torque constraints**: The three joint kinematic limits must not be exceeded. Similarly, each of the joint actuator torque limits must be respected.

**Singularity constraint**: The singularity constraint avoids the leg being completely stretched. During the thrust phase (where we assume that contact is maintained), CoM c must stay in the hemisphere of radius equal to the maximum leg extension. This condition prevents the robot from getting close to points with reduced mobility that produce high joint velocities in the inverse kinematic computation. Even though this constraint is enforced by design in the action generation, the actual trajectory might still violate it due to tracking inaccuracies. If a singular configuration is reached, the episode is interrupted and a high cost is returned. The costs caused by the violation of path constraints are evaluated for each time step of the thrust phase and accumulated into the feasibility cost $C_{f}$. In addition to these path constraints, we also want to account for the error between the actual and the desired lift-off state. This penalty $C_{lo}$ encourages lift-off configurations that are easier to track for the motion controller. Another penalty $C_{td}$ is introduced when an episode does not produce a correct touchdown. This is needed to enable in-place jumps and to prevent the robot from staying stationary. The positive component of the reward function is the output of a non-linear landing target reward function, which evaluates how close the CoM arrived to the desired target (an equivalent reward could be defined by expressing landing accuracy at the foot level because the robot is supposed to land with the same joint posture as the initial state. We expressed this at the CoM level for consistency with the trusting trajectory). This reward grows exponentially when this distance approaches zero:(9)$$R_{lt}\left(c,c_{tg}\right)=\frac{\beta }{k∥c-c_{tg}∥+\epsilon },$$
where *k* is a gain to encourage jumps closer to the target position, and β is an adjustable parameter to bound the maximum value of $R_{lt}$ and scale it. An infinitesimal value ϵ is added at the denominator to avoid division by zero. Hence, the total reward function is:(10)$$R=1_{R^{+}}\left[R_{lt}\left(c,c_{tg}\right)-\sum_{i=0}^{n_{c}}C_{i}\right]$$
with $n_{c}=8$, and where $C_{i}$ are the previously introduced feasibility costs. We decided to perform reward shaping [37], by clamping the total reward to $R^{+}$ by mean of an indicator function $1_{R^{+}}$. This aims to promote the actions that induce constraint satisfaction.

### 3.3. Implementation Details

The training of the RL agent and the sim-to-sim validation of the learned policy were performed on top of a Gazebo simulator. Because we are considering only translational motions, we modeled a 3 Degrees of Freedom (DoFs) monopod with three passive prismatic joints attached at the base. These prismatic joints constrain the robot base’s movements to planes parallel to the ground. For the sake of simplicity, we also considered that the landing phase was under the responsibility of a different controller (e.g., see [2]). Our interest was simply in the touch-down event, which is checked by verifying that the contact force exceeds a positive threshold $f_{th}$. Therefore, the termination of the episode is determined by the occurrence of three possible conditions: execution timeout, singularity, unfeasible jump, or touch-down.

The control policy (default NN) is implemented as a neural network that takes the state as an input, and outputs the actions. The NN is a multi-layer perception with three hidden layers of sizes 256, 512, and 256 with ReLU activations between each layer, and with tanh output activation to map the output between −1 and 1. A low-level PD plus gravity compensation controller generates the torques that are sent to the Gazebo simulator at 1 kHz. The joint reference positions at the lift-off are reset to the initial configuration $q_{0}$ to enable the natural retraction of the leg and avoid stumbling. Landing locations at different heights are achieved by making a 5 × 5 cm platform appear at the desired landing location only after the apex moment (making the platform appear only at the apex is needed for purely vertical jumps because it avoids impacts with the platform during the trusting phase). The impact of the foot with the platform determines the touch-down event and the consequent termination of the episode. A tight interaction with the simulation environment is key to the efficient training of the RL agent. The communication between the planner component and the Gazebo simulator is managed by the Locosim framework [38]. To interact with the planner, and consequentially with the environment, we developed an ROS node called *Jumpleg Agent*, where we implemented the RL agent. The code is available at (source code available at https://github.com/mfocchi/rl_pipeline) (accessed on 25 July 2024). During the initial stage of the training process, the action is randomly generated across $N_{exp}$ episodes to allow for an initial broad exploration of the action space.

## 4. Simulation Results

In this section, we discuss some simulation results that validate the proposed approach and compare it with state-of-the-art alternatives. We used a computer with the following hardware specifications: CPU AMD Ryzen 5 3600 (Santa Clara, CA, USA), GPU Nvidia GTX1650 4 GB (Santa Clara, CA, USA), RAM 16 GB DDR4 3200 MHz. During training, we generated targets inside a predefined training region. These samples are generated randomly inside a cylinder centered on the robot’s initial CoM, with a radius in [0, 0.65] m and a height in [0.25, 0.5] m (see Figure 3). The size was selected to push the system to its performance limits. The parameters of the robot, controller, and simulation are presented in Table 1.

### 4.1. Non-Linear Trajectory Optimization

The first approach we compare with is a standard optimal control strategy based on Feasible Differential Dynamic Programming (FDDP). FDDP is an efficient algorithm for whole-body control [39] that exploits the sparse structure of the optimal control problem. The FDDP solver is implemented with the optimal control library Crocoddyl [40] and uses the library Pinocchio [41] to enable fast computations of costs, dynamics, and their derivatives.

For the problem at hand, we discretized the trajectory into *N* successive knots with a timestep dT. As decision variables, we chose the joint torques. We split the jump into three phases: thrusting, flying, and landed. The constraints in FDDP are encoded as soft penalties in the cost. We encoded friction cones and the tracking of foot locations and velocities at the thrusting/landed stages, respectively. During the flying phase, we added a tracking cost for the CoM reference to encourage the robot to lift off. We regularized control inputs and states throughout the horizon.

### 4.2. End-to-End RL

At the opposite end of the spectrum, there are the approaches entirely based on Deep Learning, using end-to-end RL without injecting any prior domain knowledge. The RL agent sets joint position references to a low-level PD + gravity compensation controller. The use of a PD controller allows the system to inherit the stability properties of the feedback controller, but at the same time, it allows for explosive torques (by intentionally regulating the references to have a discrepancy w.r.t. the actual positions). Contrary to GRL, E2E RL queries the action at each control loop until touch down, because set points are also needed for the joints when the leg is airborne. To be more specific, instead of directly setting the joint references $q^{d}$, the control policy produces as action a and joint angle deviations $\tilde{q}\in R^{3}$ w.r.t. to the nominal joint angle configuration $q_{0}$. As suggested by [42,43], to ease the learning, we run the agent at a lower frequency (200 Hz) than the controller (1 kHz). We terminate the episode if: (1) a touchdown is detected; (2) the robot has fallen (i.e., the base link is close to the ground); (3) we reach singularity; or (4) a timeout of 2 s is reached. CoM, joint position, and joint velocities are included in the state. Since, in the E2E implementation, the joint velocity assumes non-zero values, it must be included in the state. Additionally, joint velocity has a temporal relation, which is likely the reason its inclusion in the state space improves learning performance [31]. Since the domain is not changing and the state is Markovian, augmenting the state with the history of some past samples [13,25] was not necessary. Hence, the observation is (c, q, $\dot{c}$, $\dot{q}$, $c_{tg}$). Differently from GRL, the initial state of each episode is sampled from a uniform distribution in the neighborhood of the nominal joint pose ($c_{0},q_{0}$). We also encourage smoothness by penalising the quantity ${\tilde{q}}^{j}-{\tilde{q}}^{j-1}$. Finally, because of the different units, we scale each state variable against its range in order to have better conditioning in the gradient of the NN. In terms of rewards, we use the same feasibility rewards used for GRL, computed at each loop. To discourage the robot from taking a sequence of rapid and small in-place jumps toward the target, we applied, before the lift-off, a penalty each time that the leg performs a ground contact that is not a unique touchdown. Another penalty is added to force the leg to stay still during the flying phase. The same target function (Equation 9) of GRL is used to encourage landing close to the target. Providing this reward at every step also results in a more informative (i.e., dense) reward [44] before the touchdown. Following the curriculum learning idea [45], we gradually increase the difficulty of the jump, enlarging the bounds of the training region (where the targets are sampled) in accordance with the number of episodes and the average reward.

### 4.3. Policy Performance: The Feasibility Region

We tested the agent in inference mode, for omnidirectional jumps at 6 different heights, for a total of 726 target positions uniformly spaced on a grid (test region) of the same cylindrical shape as the training region. The area of the test region was chosen to be 20% bigger than the training region, to demonstrate the generalization abilities of the system at its feasibility boundary. The policy was periodically evaluated on the test region set to assess the evolution of the models stored during the training phase. To measure the quality of a jump, we used the RPE, which we define as the distance between the touch-down and the desired target point, divided by the jump length. The RPE metric is defined as:$$RPE=\frac{∥c_{tg}-c_{f}∥}{∥c_{tg}-c_{0}∥}$$
where $c_{f}$ is the reached landing position. The feasibility region represents the area where the agent can execute an accurate landing, i.e., RPE ≤10%. A color bar quantifies the jump accuracy.

#### 4.3.1. Performance Baseline: Trajectory Optimization

We compared the approach with the baseline FDDP approach, without changing the cost weights and limiting the number of iterations to 500. For optimal control, setting dT=2 ms, the average computation time was 2.13 s for front jumps and 2.35 s for back jumps, while a single evaluation of the NN requires only 0.7 ms. Figure 4 (last 2 plots in the row) shows that a reasonable accuracy is obtained for target locations in front of the robot, while FDDP behaves poorly for targets behind the robot. We also tried different timesteps dT, ranging from 1 to 3 ms. The solver did not converge for the majority of targets when using dt>2.2 ms. Decreasing dT from 2 ms to 1 ms resulted in an increased computation time (from 2.3 to 4.3 s), while the number of feasible points increased from 118 to 168. We can easily explain the different performances for forward and backward jumps: when executing backward jumps FDDP cannot generalize owing to the asymmetry of the leg, and the solver remains trapped in a poor local minimum. We conjecture that this could be related to the leg morphology (with all knees bent backwards), which influences the main axis force ellipsoid, and hence the capability of the leg to push. These issues may be mitigated by duly tuning the weights for each target and/or augmenting the decision variables with the timings of the jump phases (at the price of a significant computational burden). Computing the mean RPE separately for the back region and the front region, we obtained an accuracy of 52 % and 16.5%, respectively.

#### 4.3.2. Performance of GRL

We repeated the simulations on the test region using GRL with the default NN and with an NN where we halved the number of neurons in each hidden layer (half NN). Figure 5 shows that, in both cases, the RPE decreases (i.e., accuracy increases) monotonically with the number of episodes. For front jumps, the average RPE goes from approximately 40% to 16% in 100 k episodes. A satisfactory level (i.e., RPE 20% for front jumps) is already achieved after 50 k episodes. All the feasibility constraints turned out to be mostly satisfied after 10 k episodes. The number of feasible points (with RPE ≤10%) after 100 k is 316 out of 726. The fact the feasible points do not cover the whole test region is an indicator of the maximum capability of the robot. The figure also shows that GRL always outperforms the standard optimization method in terms of accuracy for back jumps, achieving a comparable accuracy for front jumps.

Halving the number of neurons, the two models behave with similar accuracy, showing that the RL model could ideally be further simplified. From Figure 3 and Figure 4, we can observe that the feasibility region expands with the number of training epochs. In the same figures, we can see that the test region (black cylinder) is bigger than the region where we trained the NN (gray cylinder), demonstrating its extrapolation capabilities.

GRL is also capable, to some extent, of learning the dynamics and compensating the tracking inaccuracies of the underlying low-level controller. This is shown in the accompanying video (link to the accompanying video (https://youtu.be/ARhoYwIrkU0?si=-dchquJwMFBNd2KN) (accessed on 25 July 2024)) by the grey ball that represents the ideal landing location (i.e., if the CoM lift-off velocity associated to each action was perfectly tracked). The grey ball is different from the desired target (blue ball) because of small tracking inaccuracies but the agent learned to provide a lift-off velocity that compensates for these, managing to accurately reach the desired location (blue ball). In the same video, we show how the quality of the jumps steadily improves with the number of training episodes.

#### 4.3.3. Performance Baseline: E2E RL

The E2E policy was trained over 10M simulation steps, not reaching convergence and, consequently, not achieving satisfactory results. In the test phase, there were only a few targets out of 726 where the algorithm managed to attain an error below 10%. We are aware that E2E is capable of achieving walking tasks. However, for jumping tasks, while additional engineering effort in reward shaping [42] is believed to improve the learning speed and the performance of the E2E policy, our method provides a data-efficient alternative that is robust to the controller’s inaccuracy, generalizes to different target positions, and can be applied to different robot morphologies. More details are reported in the accompanying video.

## 5. Conclusions

In this work, we proposed a guided RL-based strategy to perform an omni-directional 3D jump on an elevated terrain with a legged robot. Taking inspiration from nature and exploiting a few heuristic assumptions on the the shape of the jump trajectory (i.e., we added some structure in the policy), we have shown that, in a few thousand training episodes, the agent obtains the ability to jump in a big area, maintaining high accuracy while reaching the boundary of its performance (coming from the physical limitations of the machine). The approach learns a robust policy and proficiently compensates for the tracking inaccuracies of the low-level controller. The proposed approach is very efficient (it requires a small number of training episodes to reach a good performance), it achieves a good generalization (e.g., by executing jumps in a region 20% larger than the one used for training), and it outperforms a standard end-to-end RL that was unable to learn the jumping motion. Compared to optimal control, GRL (1) achieved the same level of performance both in front jumps and back jumps (optimal control did not) and (2) required several orders of magnitude lower computation time.

In our future research, we plan to extend the approach to a full quadruped robot, considering also angular motions. The generalization to angular motions could lead us to a comprehensive a framework for the execution of a large class of jumping motions (e.g., twist, somersault, barrel jumps) on flat, elevated, or inclined surfaces. For a successful implementation on the real platform, a landing strategy [2] will be also integrated. We are also seeking ways to improve robustness by including robot non-idealities in the learning phase and to further speed up the training phase by leveraging parallel computation. Additionally, we plan to create a jump reflex that can be triggered multiple times, including when the robot is already in motion (e.g., non-zero initial velocity).

## Figures and Tables

**Figure 1 sensors-24-04981-f001:**
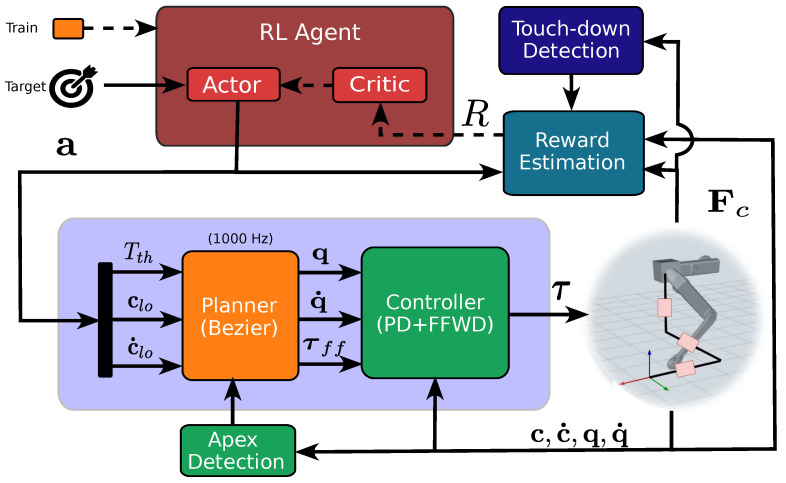
Diagram of the GRL Framework. The framework is split into two levels: the RL agent and the planner. The RL agent produces an action for the planner based on a desired target. This computes a Bézier reference curve that is mapped into joint motion via inverse kinematics and tracked by the PD controller that provides the joint torques to feed the robot. During the training, at the end of each episode, a reward is computed and fed back to the RL agent. Dashed lines are active when the framework is in training mode.

**Figure 2 sensors-24-04981-f002:**
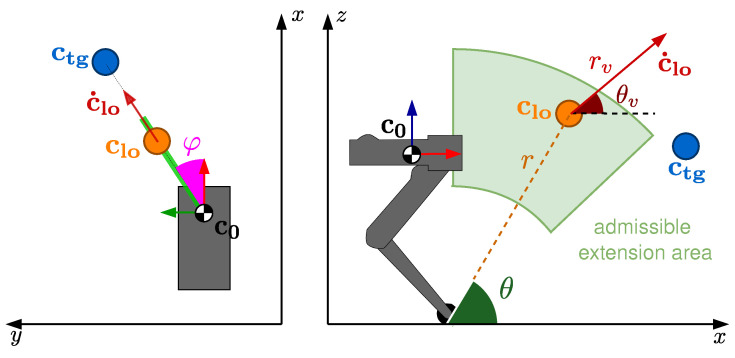
Action parametrization and its bounds. On the left is the top view, and on the right is the side view of the jumping plane.

**Figure 3 sensors-24-04981-f003:**
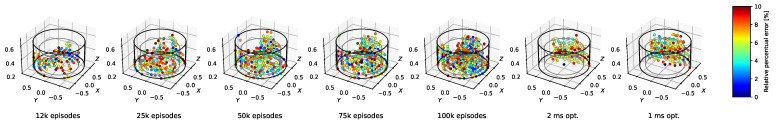
Three-dimensional view of the calculated feasibility regions (target positions with Relative Percentual Error (RPE) ≤10%). In light gray, we depict the training region bounds while in black the test region bounds. The targets are located at different heights all around the robot.

**Figure 4 sensors-24-04981-f004:**
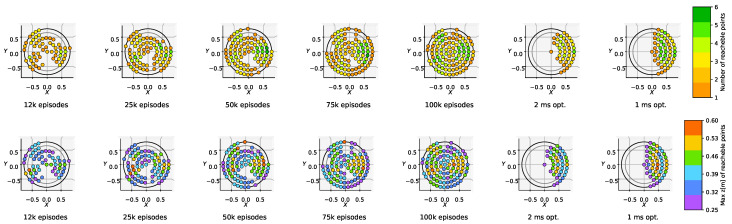
Top-view of the feasibility region: (first 5 plots in the row) for different numbers of episodes of the training phase (the number of reachable points is computed for each *X*,*Y* pair and for all the 6 different heights) and (last 2 plots in the row) in the case of the baseline FDDP.

**Figure 5 sensors-24-04981-f005:**
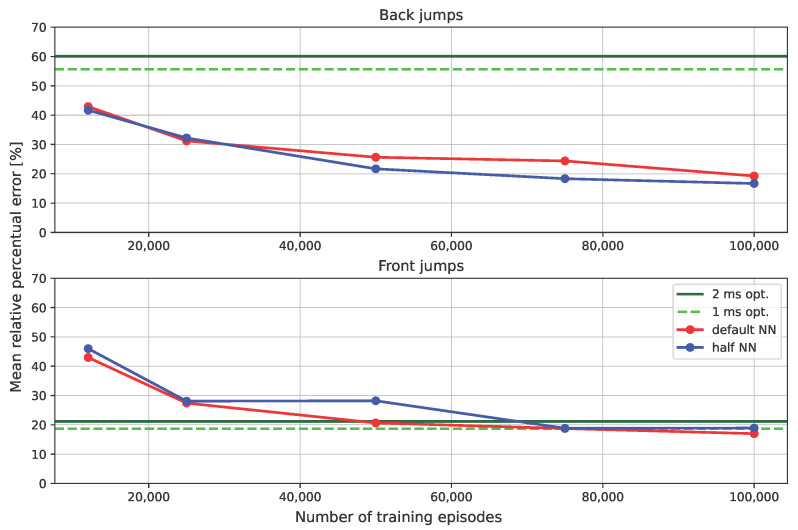
Plot of the average RPE as a function of the number of training episodes. The red plot is the NN with a default number of neurons, the blue plot is the NN with half neurons, and the green and green dashed plots are the results of optimization with 2 ms and 1 ms discretization, respectively.

**Table 1 sensors-24-04981-t001:** Controller, planner, and simulator parameters.

Variable	Name	Range
*m*	Robot mass [kg]	1.5
P	Proportional gain	10
D	Derivative gain	0.2
$q_{0}$	Nominal configuration	$\left[\begin{array}{ccc} 0 & -0.75 & 1.5 \end{array}\right]$ [rad]
dT	Simulator time step [s]	0.001
$\tau_{max}$	Max torque [Nm]	8
$f_{th}$	Touch-down force th. [N]	1
$N_{exp}$	Num. of expl. steps	1280 (GRL), 10 × 10^4^ (E2E)
$b_{s}$	Batch size	256 (GRL), 512 (E2E)
$n_{exp}$	Expl. noise	0.4 (GRL), 0.3 (E2E)
$tg_{rep}$	Landing target repetition	5 (GRL), 20 (E2E)
$N_{train}$	Training step interval	1 (GRL), 100 (E2E)

## Data Availability

Data is contained within the article and Appendix A.

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
