# Peer review of "Efficient Reinforcement Learning for 3D Jumping Monopods"

_sensors, 2024, doi:10.3390/s24154981_

Round 1

Reviewer 1 Report

Comments and Suggestions for Authors

The paper proposes a reinforcement learning method for 3D Jumping Monopods. Generally, the topic is interesting and worth study. Here are some of my comments:

1. How to choose the PD parameters. Will different parameters affect the performance?

2. An action with few parameters reduces the exploration space. Is there any lost with fewer parameters? Please discuss.

3. How to get (6) according to (5). 

4. FDDP is proposed in 2018. Please compare with some recent results.

5. In the simulation, how do authors choose parameters in the reward function (10).

Author Response

see attached pdf for detailed response

Reviewer 2 Report

Comments and Suggestions for Authors

An efficient reinforcement learning method for 3D jumping monopod robots is proposed.

(1)Although the paper proposes a novel guided reinforcement learning approach, the description in the methods section, especially regarding the neural network architecture and training details, seems to be insufficiently detailed. In order to improve the transparency and reproducibility of the paper, the authors need to provide more detailed implementation details of the algorithm, including but not limited to the configuration of the network layers, the choice of the activation function, the definition of the loss function, and the hyperparameters used in the training process. This paper presents an efficient reinforcement learning method for 3D jumping monopod robots.

(2)The results of the simulation experiments in the paper are encouraging, but to further validate the effectiveness and generalizability of the proposed method, it is recommended that the authors conduct the experiments in different environments and under a wider range of conditions. In addition, if possible, it is recommended that the experiments be conducted on a real robotic platform to demonstrate the potential of the algorithm for practical applications.

Comments on the Quality of English Language

Overall, the paper uses English grammar and syntax appropriately, and no serious errors were found. However, careful proofreading may be required to avoid minor grammatical errors.

Author Response

see attached pdf for detailed response
